# Selenium Nutritional Status of Rural Residents and its Correlation with Dietary Intake Patterns in a Typical Low-Selenium Area in China

**DOI:** 10.3390/nu12123816

**Published:** 2020-12-13

**Authors:** Xiaoya Wang, Hairong Li, Linsheng Yang, Chang Kong, Jing Wang, Yingchun Li

**Affiliations:** 1Key Laboratory of Land Surface Pattern and Simulation, Institute of Geographical Sciences and Natural Resources Research, Chinese Academy of Sciences, Beijing 100101, China; wangxy.17s@igsnrr.ac.cn (X.W.); yangls@igsnrr.ac.cn (L.Y.); kongc.15s@igsnrr.ac.cn (C.K.); jingwang@ccnu.edu.cn (J.W.); 2College of Resources and Environment, University of Chinese Academy of Sciences, Beijing 100101, China; 3Key Laboratory for Geographical Process Analysis & Simulation, Research Institute of Sustainable Development, Central China Normal University, Wuhan 430079, China; 4Centre for Disease Prevention and Control of Binxian County, Xianyang 713500, China; liyingchun1353@gmail.com

**Keywords:** Binxian, low-selenium area, hair, food consumption frequency, hierarchical, correlation

## Abstract

China is recognized as a selenium-deficient country, and nutritional selenium intake has always been a concern. To clarify the current inhabitants’ selenium nutrition status and the characteristics of dietary consumption in low-selenium areas, samples of human hair and grains were collected, and food frequency questionnaires were administered in Binxian County, Shaanxi Province, a typical low-selenium area in the Loess Plateau. The subject number of the study is 85, and the age range is from 11 to 81 years, with an average of 60. The results showed that the average hair selenium content of the residents was 231.7 μg/kg, and 62.4% of the participants had levels higher than the selenium deficiency threshold (200 μg/kg). There was a significant positive correlation between the hair selenium content and the food consumption score after adjusting for rice outsourcing. Three different dietary patterns were noted according to hierarchical cluster analysis. This study provides a tool for assessing the selenium nutrition of inhabitants in low-selenium areas and has considerable significance for improving the dietary pattern of residents.

## 1. Introduction

Selenium is an essential micronutrient for humans [1]. This micronutrient plays an important role in redox homeostasis, antioxidant defense and the immune system via selenoenzymes and selenoproteins [2]. However, selenium intake is insufficient in many countries. There are 0.5–1 billion people are affected by selenium deficiency in the world [3]. China is one of 40 selenium-deficient countries reported by the World Health Organization (WHO) [4], for there is a low selenium belt in China. The formation of a low selenium belt is determined by the migration characteristics and biological characteristics of selenium in the environment. Low selenium in the soil is the basis of its formation. A low-selenium environment leads to insufficient selenium intake through the food chain [5], which causes serious health consequences, such as Keshan disease and Kashin–Beck disease (KBD) [6,7,8]. KBD mainly occurs in children aged 5–13 years. The clinical manifestations of it are rheumatoid arthritis, shortening of fingers and toes, thickening of joints or growth disorders of organisms [3]. The geographical distribution of KBD coincides with the severe selenium deficiency belt from northeast to southwest China [9].

Since the 1980s, various comprehensive measures were taken to prevent and control Keshan disease and KBD, in addition, the socio-economic status was continuous developed and the living conditions has improvement in China, the selenium nutrition of the population has obviously increased in the low-selenium areas of China. Many KBD-affected areas have reached the national control standard, but the progression varies greatly [10]. There is still a large gap between daily selenium intake and selenium physiological nutrition requirements in rural residents living in low-selenium belt areas [11]. As diet is the crucial determinant of daily selenium intake, it is necessary to conduct a study on selenium nutrition status, dietary selenium intake pathways and daily selenium intake amount to explore the effect of the evolution of food consumption characteristics on the changes in selenium nutrition among residents in typical low-selenium belt areas.

The hair selenium content is a commonly used index for evaluating human selenium nutrition as it was found that selenium levels measured in human hair have a good correction with those in kidney, liver, lung, plasma [12,13], and blood [14]. Hair selenium can be sensitive to selenium intake and in vivo selenium levels over a period of time [15,16]; therefore, hair is usually used in epidemiological studies to assess the long-term selenium nutritional status of the population, particularly for those with stable dietary habits [17,18].

To assess residents’ intake of selenium from various foods, a food frequency questionnaire (FFQ) and 3-day weighed food record (WFR) were administered. Previous studies have proven that the results of the FFQ have no significant difference from those of the WFR in assessing dietary energy and nutrient intake [19,20]. The FFQ focuses on the overall dietary structure and can reflect nutrient intake patterns [21,22], and it is of guiding significance for dietary improvements. However, due to the large variation in soil selenium, the selenium intake from crops in different regions differs greatly. Therefore, the food composition table is not suitable for estimating selenium intake, which should be instead of the actual value of selenium in local crops [2].

Binxian County, located in mid-western Shaanxi Province on the south of the Loess Plateau in China, had overt deficiencies in selenium nutrition levels and a serious KBD prevalence in the 1980s [23]. The child incidence of KBD was 43.33%, and the average hair selenium content of the population was only 140 μg/kg at that time [24,25]. Previous studies showed that the dietary structure of the population of Binxian County was unbalanced in the 1980s, the main staple food of the residents was local self-grown wheat; there was a greater consumption of pickles than of stir-fry; and the consumption of meat, milk, eggs and fruit was very low. With economic development and a series of measures to prevent and control KBD [10,26], residents’ living standards have been improving gradually, selenium nutrition has also improved to a certain extent, and the risk of KBD in Binxian County has been greatly reduced. However, there has been no systematic assessment of whether the effects of a low-selenium environment have been removed to date or of the relationship of these effects with changes in the food consumption structure of rural residents.

In this study, Binxian County was selected as a typical study area in which to explore the following two questions: (1) Does the current nutrition status of selenium meet the physiological selenium requirement for rural inhabitants in a typical low-selenium area? (2) What are the dietary habits of the inhabitants, and what is the comprehensive relationship between selenium nutriton levels and food consumption characteristics? This work provides a scientific basis for proposing relevant nutrition strategies and guiding the improvement of selenium nutrition in residents living in selenium-deficient areas.

## 2. Materials and Methods

### 2.1. Materials Sampling and Pretreatment

The study protocol was approved by the Center for Disease Prevention and Control of Binxian County and the agreement of the village clinics. All individuals gave their informed consent for inclusion before they participated in the survey. According to the distribution of KBD endemic areas in Binxian County and the terrain features of this area, 11 villages were selected as the sample sites. Hair samples from 5–10 villagers were collected with stainless steel scissors, and grain samples were also collected. Additionally, a food frequency questionnaire was administered in each selected village. Rice samples were obtained from a local market. Ultimately, we obtained 85 hair samples from 42 men and 43 women with an average age of 60 years, including 38 people diagnosed with KBD according to the clinical diagnostic criteria and 47 people without KBD. We obtained 66 valid questionnaires, 66 wheat samples and 4 rice samples. Figure 1 presents the sampling sites in Binxian County.

Hair samples of no less than 2 g were collected from pillows and placed in a clean glass beaker, soaked in a specific proportion of neutral detergent solution for 24 h, stirred several times with a glass rod during immersion, and then successively washed with tap water, distilled water and ultra-pure water. The hair was then heated at 60 °C for approximately 4 h in an oven. Then, the dried hair was cut into approximately 5-mm pieces with ceramic scissors and stored for analysis.

The non-powdered grains were successively washed with tap water, distilled water and ultra-pure water, dried at a low-temperature (not higher than 60 °C) in an oven, crushed in a stainless steel mill (FW-100, Taisite Instrument Co., Ltd., Tianjin, China) and then bagged for use [27]. The flour was immediately bagged for use.

### 2.2. Calculation Methods

The food questionnaire examined the rural residents’ frequency of consumption of various foods in the past month. According to the statistical yearbook data for Shaanxi Province, the consumption of aquatic products by rural residents in Shaanxi Province was only 1.0 kg/year in 2017 [28], which had little impact on selenium intake. Moreover, the food consumption frequency pre-investigation indicated that local residents rarely ate aquatic products; hence, the consumption of aquatic products was not considered. All foods were divided into eight items, namely: grains, potatoes, beans, eggs, meats, milk, vegetables and fruits, on the basis of the Food Guide Pagoda for Chinese residents. The daily consumption frequency score for each food item was calculated using the method shown in Table 1. The frequency scores for each item were added to yield the total food consumption frequency score (FFQS). The formula is shown below as Formula (1).
(1)$$\mathrm{FFQS}\;(\mathrm{food}\;\mathrm{frequency}\;\mathrm{questionnaire}\;\mathrm{score})=\sum_{i=1}^{i=8}frequency\;score$$

The second step was to consider the effect of exogenous foods. In addition to the relatively large amount of local wheat, outsourced rice was found to be an additional staple food for some residents in the study area. The selenium content of the outsourced rice was obviously higher than that of the local wheat grown in the low-selenium environment [30]. The calculation divided the selenium content of the rice by the selenium content of the wheat and then multiplied the resulting value by the rice frequency score. This accounted for the extra intake of selenium from rice consumption compared to that of wheat consumption. The obtained value was included in the grain score in the second step. The following Formula (2) shows the calculation method:(2)$$\mathrm{FFQS}\prime =\sum_{i=1}\left(1+\frac{rice\;selenium\;content}{wheat\;selenium\;content}\right)\times frequency\;score+\sum_{i=2}^{i=8}frequency\;score$$

The third step was to evaluate the residents’ daily selenium intake amount. This method multiplied the adjusted score in step 2 by the selenium content and consumption amount of all 8 food items. The selenium content data for each item except for grains were derived from the literature, the Chinese Food Composition Tables (Standard Edition) and The Fifth China Total Diet Study [30,31,32]. The consumption data were obtained from the Food Guide Pagoda from the dietary guidelines for Chinese residents [33]. The median interval value of the recommended intake per food item was used. Table 2 contains the details for these data. An individual daily selenium intake amount was calculated using Formula (3):(3)$$\mathrm{SIA}\;(\mathrm{selenium}\;\mathrm{intake}\;\mathrm{amount})(\mu g/d)=\sum_{i=1}FFQS^{\prime }\times wheat\;selenium\;content\times consumption+\sum_{i=2}^{i=8}FFQS^{\prime }\times \;selenium\;content\;of\;food\;item\times consumption$$

### 2.3. Chemical Analysis

The determination of total Se referred to national standards for the determination of Se in foods (GB/T 5009.93–2003). Hair and grain samples were digested in a mixture of concentrated HNO_3_ and H_2_O_2_ (V1:V2 = 2:1) on a hot plate at below 170 °C until the solution became clear, and then the selenium content was determined by inductively coupled plasma mass spectrometry (ICP-MS) [34]. Quality control was carried out according to the national standard materials GBW 07601a (human hair), GBW 10010 (rice) and GBW 10011 (wheat) for composition analysis, and 10% of the samples were analyzed repeatedly. All the chemical reagents used were guaranteed to be of analytical grade.

### 2.4. Statistical Analysis

Data processing and chart production were mainly performed using SPSS 19.0 and Microsoft Excel 2010. A *t*-test was used to compare the differences between participangs diagnosed with KBD and those who were not. The Kruskal–Wallis H test was used for multiple independent sample comparisons. The hierarchical cluster method was used to analyze the characteristics of the residents’ diet. The Spearman correlation coefficient was applied in this study, and the difference was significant at the 0.05 level.

## 3. Results

### 3.1. Selenium Content in Hair

The average hair selenium content of the study area is shown in Figure 2. The results show that the average hair selenium content of the residents was 231.7 μg/kg, which was higher than the threshold for selenium deficiency (200 μg/kg) in KBD endemic areas [35]. Compared with the hair selenium content of other areas in Shaanxi Province, inhabitants living in the study area had a lower hair selenium content than those of adjacent Linyou County (370 μg/kg) and Yongshou County (400 μg/kg) [36]. As a contrast, the average hair selenium content of Shaanxi Province was 290 μg/kg, the same as Heilongjiang Province, while in Guangxi Province is 390 μg/kg [11]. There were 38 patients with KBD, and they exhibited an average hair selenium content of 223.6 μg/kg, ranging from 66.3 μg/kg to 381.2 μg/kg; the 47 healthy participants had an average hair selenium content of 238.3 μg/kg, ranging from 63.9 to 648.5 μg/kg, and there was no significant difference between the two groups (*t* = 0.878, *p* > 0.05). Figure 2 shows that 32 participants had hair selenium content below 200 μg/kg, which accounted for 37.6% of the total participants, and the number of patients in females is slightly higher than in males; 19 participants had selenium content between 200 μg/kg and 250 μg/kg, accounting for 22.3% of the total; 33 participants had moderate hair selenium levels, accounting for 38.8% of the total participants; and 1 female participant had a selenium content higher than 500 μg/kg.

### 3.2. Correlation between Hair Selenium Content and Food Consumption Characteristics

According to the calculated results, the FFQS ranged from 1.16 to 6.80, and the coefficient of the correlation between the FFQS and hair selenium content was 0.250 (*p* < 0.05). The FFQS’, which is the FFQS adjusted for the ratio of the selenium contents of rice and wheat, ranged from 1.30 to 6.85, and the coefficient of its correlation with hair selenium content increased to 0.518 (*p* < 0.01). This showed a good correlation between the FFQS’ and the hair selenium content. Figure 3 presents the scatter diagram for the FFQS, FFQS’ and hair selenium content. As shown in Figure 3, after the initial FFQS was adjusted for the ratio of the selenium contents of rice and wheat, the correlation coefficient increased significantly and linearly. Therefore, the FFQS’ was more suitable for the assessment of selenium nutrition in residents of low-selenium areas than the FFQS was.

Based on the recommended consumption of all kinds of foods in the Food Guide Pagoda for Chinese residents, the selenium content of local crops and other food selenium content data obtained from food composition tables, the residents’ daily selenium intake amount (SIA) was determined to range from 5.10 μg/d to 41.32 μg/d, and the coefficient of the correlation between SIA and the residents’ hair selenium content was 0.190 (*p* > 0.05). This relatively low correlation coefficient shows that SIA is not a suitable method for evaluating individuals’ daily selenium intake in a low-selenium area.

### 3.3. Hierarchical Cluster Analysis of the FFQS’

Based on the high correlation between FFQS’ and hair selenium content, hierarchical cluster analysis was applied to the food consumption score (FFQS’) calculated in step 2 for all 8 food items. Summarizing the food consumption score of these three clusters, the average score of the 8 items food is shown in Table 3. The average hair selenium content and the average FFQS’ corresponding to the three clusters are outlined in Table 4. The average hair selenium content of cluster 1 was 200.19 μg/kg; for cluster 2, it was 231.99 μg/kg; and for cluster 3, it was 288.50 μg/kg. There were significant differences in hair selenium content among the three clusters (*χ**^2^* = 10.625, *p* < 0.05). Moreover, clusters 1 and 3 (*t* = −3.884, *p* < 0.01) and clusters 2 and 3 (*t* = −2.045, *p* < 0.05) had significant differences in terms of hair selenium content. The corresponding average FFQS’ of clusters 1, 2, and 3 was 2.72, 4.12 and 5.90, respectively, and there were significant differences among the groups (*p* < 0.01).

Specifically, there were 35 participants in the first food consumption pattern group (cluster 1), accounting for 53.03% of the total. The average hair selenium content in cluster 1 corresponded to the threshold for selenium deficiency (200 μg/kg). Cluster 1 had the largest number of individuals, which reflected the main dietary habits of the local inhabitants with a limited intake of outsourced food, beans, meats, milk and fruits. This shows that there was a certain degree of dietary imbalance in the population, which is consistent with other studies in China [37]. In comparison, the residents in the second and third food consumption clusters (cluster 2, cluster 3) had relatively abundant diets, and their selenium nutrition content were also higher than those of the residents in cluster 1. In particular, although the average consumption frequency score for grain* was slightly lower in cluster 3 than in cluster 2, the average consumption of the other daily food items was significantly higher in cluster 3 than in the other two clusters, and the hair selenium content was also exceeded the healthy threshold (hair selenium ≥ 250 μg/kg).

## 4. Discussion

In China, the economy has developed rapidly, and the income of farmers has continued to grow since the reform and opening up. Therefore, the living standard of residents has improved significantly, and food diversity increased. With the comprehensive prevention and control measures such as returning farmland to forest and grassland, eating selenium-added salt, the selenium nutrition level of the residents was significantly improved. Thus, many KBD patients have reached the selenium deficiency threshold. The selenium content in the hair of 62.4% of the residents were higher than the selenium deficiency threshold (200 μg/kg) [23], and the selenium content in the hair of 38.8% of the population reached the moderate level of selenium nutrition. However, the overall level of selenium is still lower than that of neighboring counties. The study suggests that there is no significant difference in selenium nutrition levels among residents in the KBD-affected area, which may be because the selenium nutrition of residents is affected not only by the local low-selenium environment but also by dietary structure, family economic factors, etc.

Although the low-selenium environment still exists, the structure of the residents’ diet is gradually changing. The residents’ dependence on the local low-selenium environment was reduced by the consumption of outsourced rice with high selenium content. The residents’ selenium nutrition status could be roughly evaluated based on the composition of local and imported staples and the ratio of the selenium contents of the two staple foods.

According to the hierarchical cluster analysis, it is not difficult to find that in addition to increasing the intake of outsourced staple foods with high selenium contents, the consumption of various kinds of food also appears to be particularly important to improve selenium nutrition. Filippini also found that selenium intake was closely related to the frequency of consumption of meat, grains, milk and dairy products in a population survey of communities in northern Italy [38]. Based on the data in this study, if the FFQS’ of a resident reaches 4.12, people can be considered to no longer be selenium deficient. The FFQS’ could provide a reference for evaluating the selenium nutrition level via daily diet patterns in residents and could provide an approximate range for the selenium health threshold. Although the daily intake of selenium could be accurately estimated by investigating the intake amount of all kinds of food and determining the selenium content of various foods, doing so would require considerable effort and time. The FFQS’ established in this study can roughly estimate the selenium nutrition level of rural residents by obtaining the consumption frequency of all kinds of food and the ratio of the selenium contents of local staple foods to that of outsourced staple foods, which may have great significance for evaluating the dietary structure and selenium nutrition level in other low-selenium areas. However, a smaller sample size is a limiting factor in this study. In future studies, the sample size will be expanded in order to obtain more suitable methods.

## 5. Conclusions

In summary, although the hair selenium content of residents in Binxian County is no longer indicative of an overall selenium deficiency, some rural inhabitants are still on the verge of selenium deficiency or a potentially selenium deficient. A significant positive correlation between FFQS’ and hair selenium content was noted in this study, which indicates that the FFQS’ established in this study is a practical tool for estimating the current selenium nutrition levels of residents in low-selenium areas. This correlation can clarify the diet pattern and roughly assess the selenium nutrition level based on the consumption frequency of various non-staple foods and the ratio of locally and imported staple foods. There is practical significance in improving the selenium nutrition level and controlling the risk of endemic diseases caused by selenium deficiency by increasing the proportion of non-local high-selenium staple foods consumed and the diversity of food types consumed for the current residents.

## Figures and Tables

**Figure 1 nutrients-12-03816-f001:**
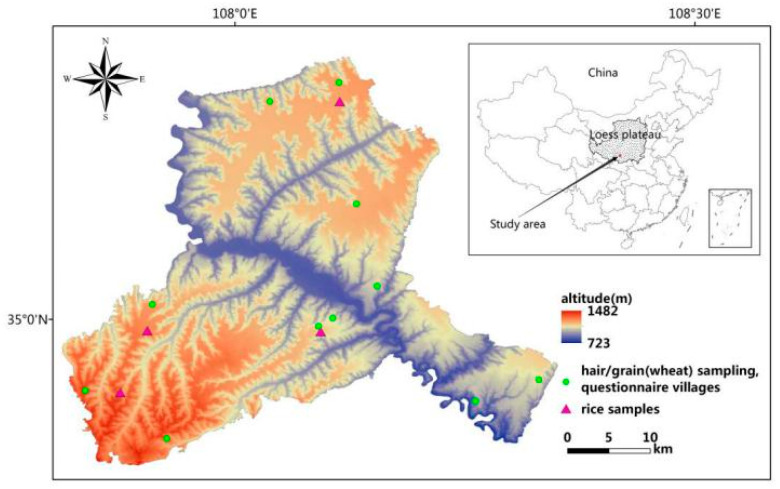
Distribution of sampling sites.

**Figure 2 nutrients-12-03816-f002:**
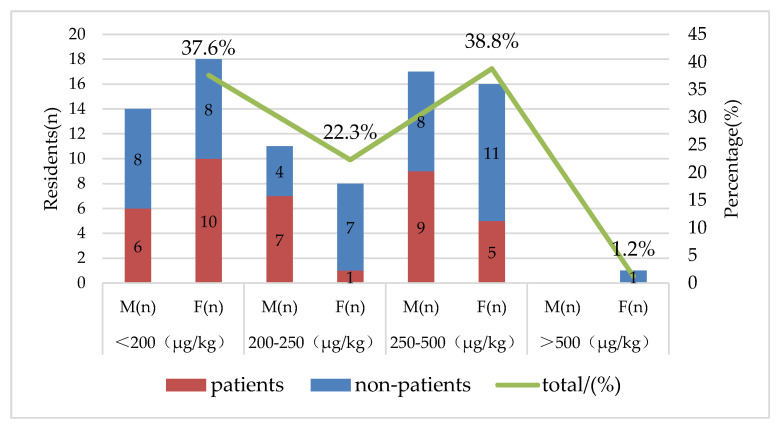
Distribution of hair selenium content in the study area (M: Male, F: Female, *n*: number).

**Figure 3 nutrients-12-03816-f003:**
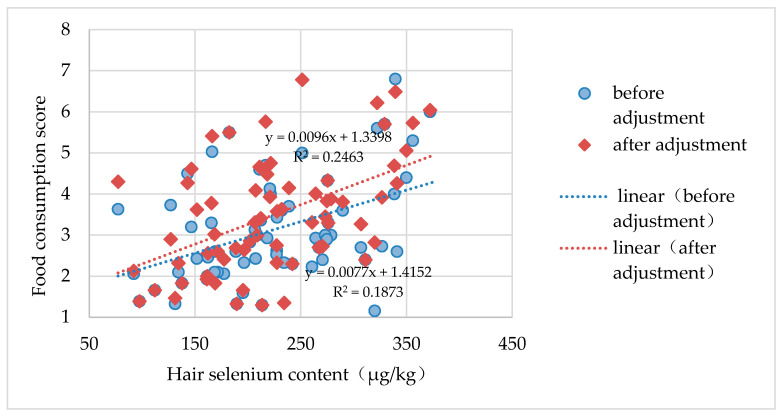
Scatter diagram of food consumption frequency score and hair selenium content (μg/kg).

**Table 1 nutrients-12-03816-t001:** Categories of consumption frequency [29].

Score	Category	Definition
1	≥1/day	Once/more than once a day
0.7	4–6 */week	4–6 times a week
0.3	1–3 */week	1–3 times a week
0.1	2–3 */month	2–3 times a month
0.03	1 */month	Once a month
0	0/rarely	Never/hardly ever

* =times (i.e.,: 4 */week = 4 times/week).

**Table 2 nutrients-12-03816-t002:** Selenium content and consumption data.

Item	Selenium Content(μg/kg)	Consumption Recommended in the Food Guide Pagoda (g)	Consumption Used in This Study(g)	Sources
Grains	Results of this study	250–400	325	/
Potatoes	14.4	50–100	75	[32]
Beans	48.8	25–30	27.5	[30]
Meats	78	40–75	60	[32]
Milk	39.1	300	300	[30]
Eggs	225.3	40–50	45	[30]
Vegetables	4.8	300–500	400	[32]
Fruits	1.62	200–350	275	[31]

**Table 3 nutrients-12-03816-t003:** The average consumption score for all 8 food items.

Cluster	*n*	Grains *	Potatoes	Beans	Meats	Milk	Eggs	Vegetables	Fruits
1	35	1.30	0.16	0.09	0.05	0.01	0.24	0.74	0.13
2	21	1.72	0.25	0.21	0.15	0.11	0.36	0.86	0.46
3	10	1.52	0.59	0.42	0.57	0.28	0.83	0.94	0.77

Grains *: after adjustment for rice and wheat.

**Table 4 nutrients-12-03816-t004:** The average hair selenium content and FFQS’ of the three clusters.

Cluster	*n*	Hair Selenium Content (μg/kg) Mean ± S.D.	FFQS’ mean ± S.D.
1	35	200.19 ± 59.27 ^a^	2.72 ± 0.91 ^a^
2	21	231.99 ± 69.49 ^a^	4.12 ± 0.50 ^b^
3	10	288.50 ± 77.08 ^b^	5.90 ± 0.58 ^c^

Two rows in the same column with different letters indicate a significant difference, while those with the same letter indicate no significant difference.

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
