# Peer review of "Selenium Nutritional Status of Rural Residents and its Correlation with Dietary Intake Patterns in a Typical Low-Selenium Area in China"

_nutrients, 2020, doi:10.3390/nu12123816_

Round 1

Reviewer 1 Report

It is an interesting issue looking at the Se nutritional status of rural residents in China. However, the number of included individuals with 85 is very low. This must be mentioned as a Limitation of the study in the discussion.

The number and age range of the study subjects must also be mentioned in the Abstract.

The conclusion must be modified, since the statement "There is Long-term siginificance..." is not supported by the data by the present study.

A similar graph as in Fig. 2 could be of interest to the reader for the resultas of KBD vs. healthy individuals. Furthermore, clinical background of KBD could be explained in the introduction, since it represents a rare disease. 

Reviewer 2 Report

The manuscript assesses the selenium status of villagers in an area of China classically known for its prevalence of Kahsin-Beck disease, a selenium-deficiency pathology. The manuscript is sound and well-written. However, there is no separation of results and discussion, which makes it rather difficult to follow the thought process involved in this manuscript. I strongly suggest to separate these two sections, plus keeping the Conclusion. After all, there is not much discussion provided, which really is where this paper could shine.

Also, the first sentence in the abstract has issues. "China is recognized as a selenium-deficient country, and nutritional selenium intake has always been a concern." Although the authors provide basis for this classification (WHO), it is well-known that not all China is selenium deficient. In fact, certain areas are the opposite, with some of the highest selenium levels in the environment and in consumption in the world. It is suggested that this sentence is edited in a more cautious tone to expresses the reality of the country in terms of selenium levels.

It is puzzling that KBD patients presented hair selenium levels above deficiency cutoff. Particularly considering the measurement of selenium deposition in the hair, which usually is a better indicator for long-term selenium exposure. What could be the rationale for this discrepancy? The authors should ponder and discuss this result in more depth.

An ethical concern is the lack of clarification regarding consent and IRB approvals as well as a more comprehensive description of the sampled population.

Minor corrections:

  • L31: "thRough"
  • L39: "evolutionary process" should be changed to "progression".
  • L72-75: there are in fact 3 questions being asked. This needs to be rectified in the text.
  • L85: "according TO the"
  • Text in item 2.4, and Results  are not in justified format.
  • Sentence in L.172 ", which may be because..." and L. 222-236 belong in a discussion, not results.

Reviewer 3 Report

This manuscript examines the cross-sectional association of selenium status with dietary patterns among 85 adults with an average age of 60 years, living in a low-selenium area in Shaanxi, China. The authors found that hair selenium levels were positively correlated with the frequency of total food consumption, and they identified 3 different dietary patterns.

My comments:

  1. It is strongly recommended that authors get editing help from someone with full professional proficiency in English, especially in the area of nutrition research. This manuscript is somewhat difficult for me to follow because the expressions in the main text are not commonly used in nutritional sciences.
  2. There is no discussion section in this manuscript, which is quite unusual for a research article. Please have separate paragraphs for results and discussion.
  3. Line 31, please specify the reasons for low selenium levels in China. In addition, specify the sources of selenium among these study participants.
  4. What was the iodine status among these people? The status of selenium may be affected by iodine deficiency.
  5. I was wondering if any participants may have health conditions that could affect the absorption of selenium, such as Crohn's disease, ulcerative colitis, kidney disorders, or other conditions?
  6. Any of these people had dyed their hair or had taken supplements of nutrients that could affect the levels of hair selenium?
  7. Specify the age range for this study. Do the authors think correlations between hair selenium levels and food consumption could differ by age?
  8. Does sex play a role in the selenium levels? What may cause higher selenium levels in females vs males?

Reviewer 4 Report

Selenium is an element that a small amount of which is needed for the proper functioning of the body. It is a component of two amino acids that they are part of enzymes important for our body. By taking the required dose of this element, we provide the body with protection against oxidative stress. Research shows that selenium may be effective in supporting the treatment of cancers, which are currently one of the greatest challenges for medicine. The effectiveness of this element was initially also observed in relation to neurodegenerative diseases and in supporting the immune response. These the optimistic results open the door to further research and improvement of therapy medicinal. It should be remembered, however, that selenium taken in excess is toxic to the body the body. May contribute directly to the development of selenosis, indirectly to development of type II diabetes, it can also induce oxidative stress in cells. 

I missed the description of the presence of selenium in various food products available in China. This message will strengthen the manuscript and interest readers. It is important in the context of the status of this element in the diet.

The authors do not write anything about the status of selenium in the world. This should be corrected. Add article to Introduction:

(2019). Selenium–fascinating microelement, properties and sources in food. Molecules24(7), 1298.

(2016). Current knowledge on the importance of selenium in food for living organisms: a review. Molecules21(5), 609.

The results are not well described. It is worth paying attention to other publications where the authors discuss the selenium status. Add comparisons with other publications. Increase the discussion of the results.

The authors should add individual legal acts that discuss the topic of selenium status in men and women.

The authors say nothing about the importance of this element in human nutrition, its antioxidant abilities, etc. It should be improved.

What about measuring selenium in human plasma? The decision to take selenium supplementation should be preceded by measurement the level of the element in the patient's plasma. In this way, we can protect patients from the effects of excessive selenium consumption. The authors should clarify this.

Round 2

Reviewer 2 Report

Ethical concern: It is puzzling why the revisions include a new co-author (W.J., which should be J.W. per author list in the beginning of the manuscript), with a new funding source, and with this new co-author being listed as having contributed in Methodology per the Contributions list, when no additional change in methods was made or requested during the revision process (comparing the 1st and revised version of manuscript).

Suggestion: An unfilled sample of the informed consent form could be translated and uploaded as Supplementary Material.

The sentence that opens the discussion should add "in China" to it.

Minor comments:

L.21 - "years OLD"

L.39 - "thRough"

L.236 - "many KBD patients HAVE"

L.240 - "selenium NUTRITION"

L.263 - "a SMALL sample size"

Format: Text in sections 3.2, 3.3 and discussion are not justified.

Reviewer 4 Report

The article is corrected, but the authors should revise all the literature again and adapt it to the journal's requirements. DOI numbers must also be added.
